# Alignment Control of Ferrite-Decorated Nanocarbon Material for 3D Printing

**DOI:** 10.3390/mi15060763

**Published:** 2024-06-06

**Authors:** Narit Boonhaijaroen, Pitchaya Sitthi-amorn, Werayut Srituravanich, Kwanrat Suanpong, Sanong Ekgasit, Somchai Pengprecha

**Affiliations:** 1Technopreneurship and Innovation Management Program, Chulalongkorn University, Bangkok 10330, Thailand; 2Faculty of Engineering, Chulalongkorn University, Bangkok 10330, Thailand; 3Faculty of Commerce and Accountancy, Chulalongkorn University, Bangkok 10330, Thailand; 4Department of Chemistry, Faculty of Science, Chulalongkorn University, Bangkok 10330, Thailand

**Keywords:** 3D printing, graphene, carbon nanotube, ferrite, anisotropy

## Abstract

This paper demonstrates the potential of anisotropic 3D printing for alignable carbon nanomaterials. The ferrite-decorated nanocarbon material was synthesized via a sodium solvation process using epichlorohydrin as the coupling agent. Employing a one-pot synthesis approach, the novel material was incorporated into a 3D photopolymer, manipulated, and printed using a low-cost microscale 3D printer, equipped with digital micromirror lithography, monitoring optics, and magnetic actuators. This technique highlights the ability to control the microstructure of 3D-printed objects with sub-micron precision for applications such as microelectrode sensors and microrobot fabrication.

## 1. Introduction

The evolution of 3D printing technology has progressed from using single materials to incorporating multiple materials, facilitating the creation of intricate parts comparable in performance to those made through traditional manufacturing methods. However [1], current technological advancements are increasingly focused on achieving precise control over the internal structure at a microscopic scale, leading to enhanced utilization of material performance. An emerging frontier in additive manufacturing is anisotropic 3D printing, which involves controlling the alignment of microstructures within the printed object. 

Building on the concept of anisotropy, “multiscale 3D printing” [2,3,4,5] is a method aimed at achieving optimal performance through the design and manipulation of internal structures. This approach extends the overall performance beyond the inherent properties of the base materials. Traditionally, 3D printers operate within a single scale of resolution. For example, a stereolithography (SLA) 3D printer might operate at a ten-micron scale, unable to simultaneously manipulate sub-micron features while printing at the base micron scale. Therefore, the ability to manipulate the orientation and curative embodiment of additive materials across scales, such as graphene ribbons measuring a few nanometers, can significantly enhance product performance. This approach also reduces material usage by optimizing internal structures for dedicated orientations. For instance, polymer additives aligned in one direction can theoretically achieve 63.6% higher tensile strength than those with random alignment, as shown in Appendix A. In practice, this enhancement has been reported to be substantial, such as in the case of a multi-walled carbon nanotube (MWCNT) [6,7]. Thus, multiscale manipulation, providing detailed information for fabricating objects across scales, can reduce material consumption and consequentially promote a greener manufacturing approach.

Multiscale 3D printing also involves the precise positioning and orientation of additive particles. This technique extends to numerous applications, including electrodes for microsensors and microelectronics [8,9], enabling the fabrication of sophisticated devices unattainable through traditional microfabrication. This innovation is transforming manufacturing industries and supply chains. Consider the fabrication of a microfluidic device in a single 3D printer: using articulate graphene wires and printed electronic circuits, which can potentially shift the paradigm, allowing entire devices to be fabricated in a single facility. This development benefits healthcare and agricultural applications by enabling mass customization [10] with generative AI, potentially producing customizable microsensors tailored to individual customer or patient requirements, thereby enhancing quality of life.

Many researchers have proposed various additives for anisotropic 3D printing. According to the concept of putting more designated fidelity into the engineered design, such as in topology optimization [11,12], a higher degree of articulation reduces material usage. Thus, additive materials with anisotropic structures should demonstrate superior performance if they can be manipulated precisely. Nanocarbon materials illustrate this well. Graphene, a 2D carbon allotrope, exhibits exceptional electrical and mechanical properties. Its molecular organization contributes to its overall performance. This embedded information, in terms of the organization of the bonding, when combined with multiscale architecture, makes graphene a good candidate material for state-of-the-art anisotropic 3D printing. 

Typically, graphene exists as a one to a few layers thick sheet of carbon atoms. Many research studies [6,7,13,14,15,16,17,18,19,20,21,22] have incorporated graphene into 3D printing resin. However, manipulating the alignment of such 2D nanomaterial sheets may not yield significant benefits in composite anisotropic 3D printing processes. Hence, a specialized form of graphene, known as a graphene nanoribbon (GNR), has been introduced to achieve highly anisotropic characteristics. This GNR could be considered a 1D carbon nanomaterial. By flattening the GNR, the intrinsic stress, which is the result of rolling allotropes’ carbon atoms into a tube, of the GNR is eliminated. Hence, GNR surpasses the mechanical performance of other 1D carbon nanomaterials, such as carbon nanotubes (CNT).

GNR can be synthesized through various methods, including starting from a monomer for precise structural control. However, these processes are often complex [23,24] and may require expensive reagents [25]. Alternatively, synthesis could begin with the flattening of carbon nanotubes into graphene sheets [26,27,28,29,30,31]. Genorio et al. [26,27] showcased the unzipping process of MWCNTs and the subsequent attachment of ferrite particles to nanocarbon materials, which presents an opportunity for creating an alignable and controllable nanocarbon additive for 3D printing resin. Their method involved a three-step synthesis, but the resulting product in resin showed undesired reactions with lone clusters of metallic iron, affecting its magnetic properties. Effective encapsulation techniques are needed to protect metallic particles.

Employing functionalization techniques to adhere magnetic particles [32,33,34,35] has demonstrated an increase in magnetic particle loading, albeit with a random distribution of particles across the surface of the MWCNTs. Following this principle, the optimal strategy for synthesizing anisotropic graphene entails deliberately generating active sites on graphene or carbon nanotubes. However, it is essential to acknowledge that the outcome may vary depending on the morphology of the materials utilized [36].

According to Genorio et al. [26,27], the final step involves grafting steric groups onto the surface of the tubes to prevent the agglomeration of intercalated stacks of graphene, using 1-iododecane as the reagent. This process allows the grafting of other haloalkane reagents. On the other hand, the crystallization of the ferrite particles can be initiated by the presence of hydroxyl or hydroxide groups. Hence, haloalkanes that have oxygen on the other side of the molecule could potentially be good candidates for these reactions, both grafting and initiating.

Halohydrin is a good candidate. However, in the sodium solvation process, hydroxyl groups tend to react with the solvated sodium, leading to undesired products, which affects grafting yield. Therefore, epichlorohydrin is a better candidate for this reaction. The synthesis of epichlorohydrin involves the closing of the epoxide ring in the final step using hydroxide as the reagent. This indicates that epichlorohydrin can potentially withstand the presence of hydroxide ions as well as high pH conditions. In the final process of ferrite particle formation, a nucleophile from the ferrite nanoparticle attacks the epoxide ring, leading to the ring opening and becoming the initial site for the formation of ferrite crystals [37,38,39,40]. Conclusively, in this work, we experimented with coupling GNR/CNT with ferrite particles using epichlorohydrin in a one-pot synthesis. Furthermore, epichlorohydrin is a high-volume manufactured chemical that can be synthesized starting from glycerol, a renewable chemical. This selection aligns with green synthesis principles and negative carbon material considerations [41].

Attaching ferrite particles to GNR and CNT surfaces results in different susceptibilities between the decorated particles and surrounding media, such as photopolymer resin in 3D printing, causing the particles to move and align in a magnetic field. This phenomenon is called magnetophoresis [42,43,44]. The direction and orientation depend on the type of magnetism. Ferromagnetic particles move towards a higher magnetic field strength, aligning directly with the magnetic source. These phenomena allow manipulation using energized electromagnets or magnets to create magnetic field gradients. Magnetic manipulation has the advantage over electrostatic manipulation because it does not involve high voltage. In electrostatic manipulation [6,45,46], high voltage is applied across the liquid resin, which could lead to a voltage breakdown and pose a danger to the operation. 

Various studies have showcased the precise control of particle orientation through the application of magnetic fields [6,14,15,16,17,18,47]. In particular, magnetophoresis has been utilized to regulate the movement and direction of particles [9,43,47,48,49,50]. Additionally, research conducted in [14,15,16,47] demonstrated the utilization of a multi-solenoid system to energize magnetic fields. However, ensuring sufficient field strength necessitates the use of higher conductivity or superconductive materials, which may not be suitable for certain applications.

Integrating magnetic manipulation into an SLA 3D printer equipped with a digital micromirror device (DMD) and affordable microscope optics [51] enables the printing of objects with controlled micro-to-nano structures. This innovative approach offers a promising solution for creating a multiscale anisotropic printing platform, facilitating the fabrication of electronics and micromachines.

In this work, we propose a novel mechanism utilizing epichlorohydrin as a coupling agent. We present a functional mechanism for precisely regulating the movement and alignment of decorated carbon nanomaterials for resin-based 3D printing applications. Using a straightforward permanent magnet setup, we demonstrate the efficacy of employing low-cost components for meticulous nanoscale fabrication processes.

## 2. Experimental

### 2.1. Synthesis of Decorated Carbon Materials

The synthesis of decorated carbon materials followed the procedure outlined in the research [26], Building upon prior work, the synthesis of magnetite particles involved the simultaneous introduction of ferric and ferrous ions into the reaction under basic conditions, along with water molecules, facilitated by ferrous chloride hexahydrate. 

As shown in Figure 1, the overall mechanism of the reaction begins with the unzipping of carbon nanotubes by a NaK alloy, with cracks being filled by electrons provided by the alkaline metal as the initial site. Subsequently, epichlorohydrin is introduced to attach to the tube. An excess of alkaline metal opens an epoxied group, leading to propagation polymerization. When the mixture of the magnetite precursor, ferric chloride, and ferrous chloride is introduced into the vessel, the remaining negatively charged oxygen reacts with the mixture, providing seeding sites for magnetite to agglomerate. Additionally, the hydroxyl group on the ferrite crystal attacks the newly formed epoxide group [37,38,39,40]. Finally, a small amount of distilled water is added to continue the ferrite particle formation. 

The distribution of particles along the surface of the tube is attributed to the interplay between electrostatic repulsion and the intrinsic energy of the conjugate bond between carbon atoms. In cases where the intrinsic energy between carbon atoms is low, as seen in single-walled carbon nanotubes (SWCNTs), the initial site distribution tends to be uniform. Conversely, in MWCNTs with higher intrinsic energy, the reaction tends to distribute along the unzipping axis of the tube.

#### 2.1.1. Materials

All the chemicals were analytical grade: MWCNT (NanoTech-Labs, Inc., Yadkinville, NC 27055, USA, C-Grade, 15 nm diameter, 100 μm length), Short-SWCNT (NANOGRAFI Co. Inc., Çankaya/Ankara, Turkey, 1.0 nm diameter, 1–3 µm length), FeCl_2_.4H_2_O (Alfa Aesar, Ward Hill, MA, USA), FeCl_3_ anhydrous (Fluka, Charlotte, NC, USA), 1,2-dimethoxyethane (DME, TCI, Chuo-ku, Tokyo, Japan), sodium metal and potassium metal (Alfa Aesar), epichlorohydrin (TCI), 1-iodotetradecane (TCI), and methanol (Merck, Darmstadt, Germany). 

For the 3D printing photopolymer, isobonyol acrylate (IBOA, EM70 Eternal Materials, Kaohsiung City, Taiwan), glycidyl methacrylate (GMA, TCI), triphenylphosphine oxide (TPO, TCI), and Bis (2,6-difluoro-3-(1-hydropyrrol-1-yl)phenyl) bis (2,6-difluoro-3-(1-hydropyrrol-1-yl) phenyl) titanocene (PI-784, BASF, Ludwigshafen, Germany) were used.

#### 2.1.2. Unzipping Process

The NaK liquid alloy was prepared by mixing 1 g of sodium metal and 2 g of potassium metal under mineral oil as a stock mixture.

The unzipping of the MWCNTs and SWCNTs was performed via a sodium solvation process, according to the previous review, with some modification and scale-down. This oxygen- and moisture-sensitive process was carried out in an inert atmosphere. A total of 16.5 mg of MWCNTs and 16.5 mg of SWCNTs was weighed and filled in each 50 mL capped bottle. Then, all the bottles were charged with 12 mL of 1,2-dimethoxyethane. A 200 µL of NaK alloy was then added to the premixed bottles by pinching the pipet tip into a drop of the NaK alloy under the mineral oil to prevent oxide contamination.

The prepared mixtures were sealed and placed in an ultrasonic bath for 15 min. Within a few minutes of sonication, the appearance of a blue color confirmed the solvation of the sodium ion complex in the DME. Additionally, a noticeable expansion of the nanotubes was observed. Subsequently, vigorous stirring was maintained for 24 h.

#### 2.1.3. Attaching Epoxy Group to the Tubes

The attachment of the epoxy group was performed by adding 154 µL of epichlorohydrin subsequently to 154 µL of 1-Iodotetradecane into each unzipped carbon-nanotube mixture. This process was carried out in a fume hood with a normal atmosphere. The cloudy mixture of NaCl salt then appeared. This process continued for 2 h.

#### 2.1.4. Attaching Magnetite Particles

To selectively bind the magnetite particles, the magnetite crystals were slowly coagulated and initialized by epoxide groups on the SWCNTs and multi-layer GNRs. The FeCl_2_:FeCl_3_-DME mixture was prepared by combining a 2:1 molar ratio of FeCl_2_ and FeCl_3_ powders. The powder was vigorously shaken, and then 100 mg of the mixed powder was added to 1 mL of DME. The solution was then added to each tube. The reaction was allowed to stand for 1 h. 

#### 2.1.5. Preparation for Analysis

Next, 1 mL of distilled water was added to quench and provide a hydroxy group for magnetite particle formation. The stirring was continued for 1 h, and then 100 µL of each sample was washed with 4 mL of distilled water. A black agglomeration of carbon material was then sampled and pipetted on TEM grids.

#### 2.1.6. Instruments

The XRD spectrum was observed with a Bruker D8 (Bruker, Billerica, MA, USA). TEM and EDXS were observed with a JEM-2100 (JEOL, Akishima, Tokyo, Japan). The NMR analysis was observed with a JNM-ECZ500R/S1 500 MHz (JEOL).

### 2.2. 3D Printing Photopolymer Formulation

The formulation of the photopolymer involves considering factors such as compatibility with radiative wavelengths, resolution, depth of cure, and chemical compatibility. The initial challenge is to develop a base resin that prevents the precipitation of additive nanocarbon particles. Since the carbon nanomaterials are coated with an epoxy polymer, a suitable candidate is a monomer containing hydroxy or glycidyl groups. However, when using hydroxy groups, such as those in hydroxyethyl acrylate, an excess can react with the limited epoxide groups on the surface of the modified nanocarbon material. This excess hydroxy group cannot participate in further photopolymerization, negatively impacting the curing rate of the polymer matrix. In contrast, selecting monomers with glycidyl groups, such as glycidyl methacrylate (GMA), allows for additional cationic photopolymerization and better compatibility. The use of GMA also helps to reduce the agglomeration of carbon nanoparticles.

The solidification mechanism in this work is based on free radical type I photopolymerization, which is commonly used in 405 nm SLA 3D printers. Typically, triphenylphosphine oxide (TPO) is used at concentrations of 1–3% by weight. Considering the formulation of the resin, which is based on a GMA monomer and lacks any oligomer, the required photoinitiator concentration is reduced to approximately 2% by weight.

Combining photoinitiators is essential for the printing process. While TPO exhibits effective initiation, it also possesses UV-blocking properties, limiting curing depth to a few hundred micrometers. However, printing requires a deeper resin depth to allow particles to move freely and sufficient time for rotation and alignment. Thus, bis(2,6-difluoro-3-(1-hydropyrrol-1-yl) phenyl) titanocene (PI-784) was incorporated to reduce UV blocking from the TPO, leveraging its color bleaching mechanisms and higher reactivity. However, it is crucial to strike a balance, as excessive PI-784 may result in UV light bleeding, diminishing shape accuracy in printing. Moreover, PI-784 can be affected by leaked UV light from the projector; a full formulation with PI-784 may cause leaked light to gel, thereby reducing printing performance. Achieving the right balance between TPO and PI-784 is imperative for the success of the printing process.

Isobornyl acrylate (IBOA) is also added to the formulation. IBOA reduces polymerization shrinkage and increases the resin’s viscosity. The development of the photopolymer formulation aimed to determine its suitability for 3D printing purposes. By utilizing IBOA and GMA with viscosities of 7.5 mPa.s and 15.481 mPa.s, respectively, and without any oligomers, the total viscosity was kept below 10 mPa.s. This was to optimize particle movement with minimal drag force. However, a challenge persisted, as the lower viscosity resulted in rapid sedimentation, limiting the alignment time. The printing object size is limited to a few centimeters due to the high shrinkage of 11.3% of IBOA.

The specific gravity of the decorated nanocarbon particles is high due to the presence of a substantial weight ratio of iron–ferrite particles. Increasing viscosity helps delay the precipitation of nanocarbon particles. However, higher viscosity also hinders the movement of the carbon nanoparticles. Consequently, a workable formulation is presented in this study.

The formulation begins by allowing each sample of decorated carbon material to settle. Then, 100 µL of the bottom precipitate is meticulously drained using an autopipette into a new tube. Subsequently, 500 µL of GMA is added and thoroughly shaken. This procedure is repeated three times.

The base photopolymer is prepared by blending IBOA with 1% by weight TPO and 1% by weight PI-784. To obtain 300 µL of carbon nanoparticle-added photopolymer, 100 µL of the bottom precipitate is drained, and 200 µL of the base photopolymer is added and combined.

### 2.3. Microscopic 3D Printer Design 

The core principle behind microscopic 3D printer design revolves around showcasing in situ printing and monitoring capabilities. This entails the integration of DMD-based technology coupled with polarization cameras, enabling real-time observation of both particles and polymerization processes. Additionally, a magnetic manipulator has been specifically designed to facilitate real-time computerized manipulation of magnetic fields. This comprehensive approach ensures precise control and monitoring throughout the printing process.

### 2.4. Optical System

Our experimental setup draws inspiration from [51]. As depicted in Figure 2, we utilize a DMD projector, DLP3010EVM (Texas Instruments, Dallas, TX, USA) to generate the light pattern, which is then projected through a 28 mm achromatic lens (focal length 100 mm). A beamsplitter diverts a portion of the light for observation by a camera sensor. The remaining light travels through a right-angle mirror before being focused onto the substrate by a 5× long working distance objective microscope lens mounted on a motorized linear stage. To achieve UV illumination, we replaced the blue LED of the projector with a UV LED (405 nm). The observation takes place through a 100 mm lens, and then a camera sensor. The sensor we used in the experiment was a 5 MP polarization image sensor, IMX264MZR (Sony, Tokyo, Japan). We also deploy backlighting using a 650 nm 1 W red LED, with a plastic focusing lens cover.

The objective 5× lens (MLWD-5X, Newport, Irvine, CA, USA) has an effective focal length of 40 mm; the magnification of the system is f2/f1, where f2 = 100 mm and f1 = 40 mm. The pitch of the DLP mirror is 5.4 × 5.4 µm. The projecting pattern has a resolution of 2.16 µm. The pixel size of the camera is 3.4 × 3.4 µm. However, because of the pixel arrangement of the Sony polarization sensor, the effective pixel pitch is 6.8 × 6.8 µm, and the final resolution of the camera sensor is 2.72 µm.

The projector was calibrated using USAF 1951 glass slide resolution targets (Edmund). The effective resolution was observed down to 6.96 µm (group 6:2). The projector resolution and the camera resolution were confirmed, and the video capture was filmed at 23.98 fps.

### 2.5. Magnetic Actuator

Traditionally, solenoids have been employed to articulate the magnetic field. In this work, the required field strength gradient is more than 300 mT/cm. Consequently, the magnetic field needs to be as closely situated to the vessel as possible, hence the use of 2 mL polypropylene microtubes. However, conventional solenoids generating such strength also generate excessive heat, leading to turbulence in the printing vessel. Moreover, the latent hydraulic force within the resin often surpasses the magnetic manipulating force.

The level of the resin surface is adjusted to be lower than the position of the engaged magnet. This configuration allows the levitation force to act on the resin, delaying the precipitation of the nanocarbon particles.

To address these challenges, an improved design utilizes rare earth permanent magnets. Specifically, 1 × 1 cm N52 neodymium magnets are positioned on a swing arm actuated by four RC servo motors with a load capacity of 8 kg/cm. The chassis is constructed using acrylonitrile styrene acrylate polymer (PolyLite ASA), as shown in Figure 3. The use of permanent magnets necessitates high rigidity in the swing arm, especially when engaging both sides of the magnets simultaneously.

### 2.6. Particle Manipulation

Magnetophoresis is the phenomenon of magnetic particles moving or rotating in a magnetic field. When a magnetic field is applied to a system containing magnetic particles, these particles experience a force due to the magnetic field gradient. This force can cause the particles to move.

Theoretically, the linear magnetophoresis force ($F_{mag}$) and linear magnetic energy (${Ε}_{linear}$) can be described as the following equations [42,52,53,54],
$$F_{mag}=\frac{(\chi_{p}-\chi_{m})V_{p}(Β.\nabla )B}{\mu_{0}};\;\;{Ε}_{linear}=-\frac{(\chi_{p}-\chi_{m})VB^{2}}{2\mu_{0}}$$
where $V_{p}$ is the volume of each ferrite particle, $\mu_{0}$ is the free-space permeability, and $\chi_{p}$ and $\chi_{m}$ are magnetic susceptibilities of the particle and the medium. $\left(Β.\nabla \right)B$ is the magnetic flux density and applied magnetic field gradient. Therefore, we observe that the decorated tube or strip displays anisotropic interaction due to magnetic susceptibility. Magnetic susceptibility, being derived from magnetic permeability, results in a magnetic susceptibility tensor owing to the anisotropy of magnetic permeability. The decorated carbon material, being a one-dimensional material, exhibits uniaxial magnetic properties, denoted as $\chi_{\|}=\chi_{1}>\chi_{⊥}=\chi_{2}=\chi_{3}$. 

For linear translation, magnetite (Fe_3_O_4_) can be categorized as a ferrimagnetic compound [55] with a susceptibility higher than that of the resin, $\chi_{p}>\chi_{m}$. Consequently, decorated particles will migrate towards regions of higher magnetic fields. Conversely, carbon material decorated with ferrite particles exhibits a longitudinal magnetic susceptibility, $\chi_{\|}$, greater than its axial magnetic susceptibility, $\chi_{⊥}$. Given that ferrite particles are ferrimagnetic, the total magnetic energy reaches a minimum when the tube is aligned parallel to the magnetic fields. As a result, the decorated particles will rotate and align themselves in parallel with the magnetic field. The angular magnetic energy (${Ε}_{angular}$) can be described as:$${Ε}_{angular}(\theta )=\frac{(\chi_{\|}-\chi_{⊥})VB^{2}}{2\mu_{0}}{cos}^{2}\theta$$

As shown in Figure 4, the linear magnetophoresis force is the result of the difference between the magnetic particle, $\chi_{p},$ and resin susceptibility, $\chi_{m}$. A lower resin susceptibility leads to a stronger magnetophoresis force, $F_{mag}$, albeit at the expense of the overall magnetic field strength. However, the anisotropic magnetic susceptibility is determined by the magnetic susceptibility of the magnetite particles, carbon material, and the surrounding resin. Additionally, the shape of the particles plays a significant role in their behavior within the magnetic field. Cylindrical particles exhibit the highest interaction, while entangled particles experience constraints on their movement and rotation.

The particle manipulation scheme involves the activation of a permanent magnet close to the tube to generate a magnetic field. By applying this magnetic field, particles are drawn towards it, and pseudo-magnetic field rotation can be simulated by sequentially engaging the neighboring magnets in both clockwise and counterclockwise directions. In Figure 5, in the first step (**Step 1**), the NE magnet in the northeast direction is initially activated, causing the particles to move towards the northeast. In the second step (**Step 2**), a counterclockwise rotation is performed by retracting the NE magnet and simultaneously engaging the NW magnet. Subsequently, the first target region is cured and fixed to the position of Particle 1 (**Step 3**), after which the second particle (Particle 2) is manipulated by engaging the NW magnet. In the last step (**Step 4**), Particle 2 is then rotated by retracting the NW magnet and activating the SW magnet to create a counterclockwise magnetic field.

### 2.7. Printing Procedure

As depicted in Figure 6, the printing process involves three main steps. Firstly, the microtube base is prepared by adding 500 µL of photopolymer primer to the bottom, allowing any remaining bubbles to escape. A red UV-safe chessboard is projected onto the surface to calibrate the focusing plane. UV light is then gradually applied to the surface to initiate polymerization, with careful control to prevent uncontrolled exothermic reactions. During this stage, a 10–20 µm change in the focal plane may be observed due to polymerization shrinkage, necessitating recalibration of the chessboard to the new focal plane.

Secondly, 100 µL of prepared resin containing decorated carbon material is added to a prepared microtube with an inside diameter of 8.5 mm, resulting in a layer height of approximately 440 µm. The shape of the particles is observed, and the cylindrical shapes are selected. Within a limited timeframe of approximately 3–5 min before precipitation occurs, a magnetic swing arm is engaged to induce movement of the selected particle. The sequence of movement depends on the desired direction, with some oscillating movement required. 

Finally, once the selected particle is in place, UV light is projected onto the desired curing shape. The required curing time is divided into 6 sets, with 5 s of curing followed by 2 s of rest, to prevent overheating from the exothermic curing process. During this process, if the layer is thick, a sweeping focus is then enabled. The curing object can be observed by changes in the refractive index appearing on the surface of the liquid.

## 3. Results and Discussion

### 3.1. Attachment of Ferrite Nanoparticles

The dispersion of ferrite nanoparticles on the surface of the graphene/CNT was observed using a transmission electron microscope (TEM) equipped with energy-dispersive X-ray spectrometry (EDXS), which allowed for the correlation of ferrite particle mapping on the graphene/CNT. Additionally, the interference pattern of the graphene/CNT was observed, aiding in the identification of the separation between the polymer layer and the carbon material backbone.

An intriguing finding revealed morphological differences in the concentration and variation in the carbon nanotube substrates. The dispersion on the SWCNTs appeared uniformly distributed, contrasting with the MWCNTs, where the distribution of ferrite particles was observed to be agglomerated and packed on the ripped edges of the tubes.

As shown in Figure 7, the unzipped SWCNT surface depicts uniformly dispersed ferrite particles with diameters ranging from 5–10 nm. This diameter of ferrite may exhibit superparamagnetic due to its single domain. The measured diameter of bundled fibers was larger, ranging from 100–150 nm, compared to a single tube diameter of 1–2 nm. This observation indicated the bundling of multiple SWCNTs, which exhibited a higher reaction rate, resulting in evenly distributed reaction sites and rapid formation of ferrite particles compared to the polymerization of the epoxide group. The presence of coated epoxy polymer was confirmed by the disappearance of the interference region on the border of the projected tube image.

Figure 8 displays the NMR analysis of the intermediate reactivity following the introduction of epichlorohydrin to exfoliated SWCNT, which revealed the presence of grafted epoxide groups on the SWCNT surface. Small peaks corresponding to NMR shifts at 2.65 (C), 2.84 (B), and 3.20 ppm (A) were detected, corresponding to characteristic peaks of the epichlorohydrin structure [56]. Notably, the absence of the epichlorohydrin peak at 3.74 ppm suggests the departure of hydrogen at position (E), indicating the formation of bonds between the epichlorohydrin molecules and the SWCNT. It should be emphasized that the NMR signal at 3.47 ppm may be masked by adjacent peaks originating from DME at 3.545 and 3.394 ppm [57]. Regrettably, the removal of the DME is unfeasible due to its pivotal role in preventing particle agglomeration. However, an additional peak at 3.65 ppm was observed, possibly originating from positions (E) or (D).

In Figure 9, XRD peaks are observed. A SWCNT sample is used to validate the formation of ferrite particles, the polymerization of the epoxy, and the unzipping of the SWCNTs. The ferrite formation is observed at 16.70(111), 28.05(220), 34.04(311), 40.22(400), and the broad at 62.86–66.70 [58,59], which indicates the involvement of organic substrates. A thin layer of epoxy polymerization and unzipping of SWCNTs are observed at 17.00 and 8.41 with d-spacing of 5.2 Å and 10.51 Å [60].

The analysis of raw SWCNT samples showed no deposition of ferrous material, while raw MWCNT samples exhibited some ferrous clusters in the core of the material, serving as seeding sites in the MWCNT production. Additional experiments confirmed the performance of epichlorohydrin as the coupling agent by comparing it with absent conditions. Methanol was utilized as a quencher in the controlled experiments. The results for both MWCNT and SWCNT confirmed that the deposition of ferrite nanoparticles was rarely observed and did not attach well to the surface of the tubes.

Figure 10 shows the EDXS images, wherein the Fe image confirms the distribution of magnetite nanoparticles. The O image confirms the distribution of epoxy polymer along the surface of the SWCNT, which is normally absent on the raw surface of SWCNT. It is important to note that the single-walled carbon nanotubes (SWCNTs) were unzipped into thinner bundles of tubes and single-layer graphene, as evidenced by the splitting observed in the TEM image. This process resulted in reduced contrast in the carbon image. Additionally, the polymeric TEM grid, which also contains carbon, contributed to the detection of some carbon atoms.

### 3.2. Formulating Photopolymer for 3D Printing

The formulation of the photopolymer resin containing decorated ferrite particles, utilizing both decorated SWCNTs and MWCNTs, yielded distinct outcomes. Following the decoration process, as shown in Figure 11, the SWCNTs exhibited a bundled structure, whereas the MWCNTs expanded into layers of graphene nanoribbon sheets, likely due to the intrinsic stress in the multi-walled structure [36]. These graphene nanoribbon sheets tended to overlap, forming a robust network with high hydraulic drag, impeding free movement. Additionally, the formation of this graphene network reduced the overall anisotropy of the bundle, affecting both its mechanical properties and magnetism. Consequently, the decision was made to proceed with utilizing decorated SWCNTs for 3D printing demonstrations due to their ease of manipulation. 

### 3.3. 3D Printing

The 3D printing experiment was conducted and observed to demonstrate the precise control of anisotropic printing. As shown in Figure 12, two bundles of filaments (Particles 1 and 2) were manipulated to move and align into two target regions and then solidify, respectively.

Cylinder-shaped bundles were chosen for the printing process, with dimensions approximately 1–1.5 µm in diameter and 8–10 µm in length. The observed linear movement velocity was approximately 4 µm/s, while the rotational speed was 20 revolutions per second. The linear movement velocity varies depending on the initial direction of the bundle. An eclipse pattern with a size of 100 × 50 pixels, or approximately 108 µm × 54 µm, was projected onto the desired region. Polymerization was observed through changes in the refractive index. The embossed surface with trapped carbon nanomaterial was confirmed, and the orientation was as desired. This process must be completed within 5 min, as after this the bundles will become all sediment and no longer allowed to move.

Selecting a bundle that is floating higher, near the surface, is easier and provides more time to work with. This selection is achieved by using the depth of field and adjusting the focal plane of the microscope to be close to the surface. It can be concluded that the decorated carbon material is much heavier than the resin. Therefore, to extend the printing window time, a shaking or buoyancy mechanism is necessary.

## 4. Conclusions

This study focuses on the synthesis of ferrite-decorated graphene and carbon nanotubes, which can be manipulated within a magnetic field. Additionally, we have developed an alignment-capable 3D printing platform, with precise control down to micron resolution, utilizing cost-effective equipment. Our platform facilitates the accurate placement of carbon material bundles within the solidification region. Moreover, we have introduced a novel process for decorating ferrite particles and unzipping carbon nanotubes, which holds promise for grafting other substrates for sensor applications. The examination of the morphology of the decoration processes has been meticulously conducted, leading to the determination that decorated SWCNT is the optimal material for alignment-capable 3D printing. This innovative approach unlocks possibilities for various applications, including selectively depositing electrodes for electrochemical sensing.

## Figures and Tables

**Figure 1 micromachines-15-00763-f001:**
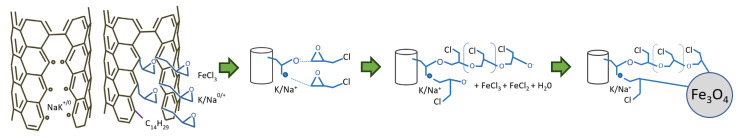
Proposed Ferrite Decoration Scheme.

**Figure 2 micromachines-15-00763-f002:**
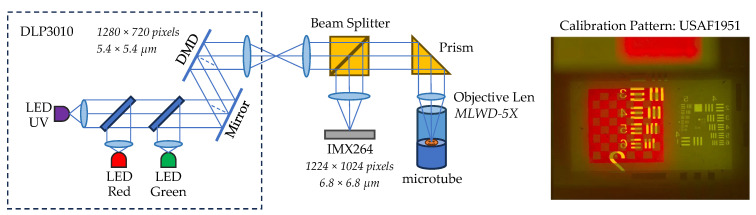
Optical Design and Construction.

**Figure 3 micromachines-15-00763-f003:**
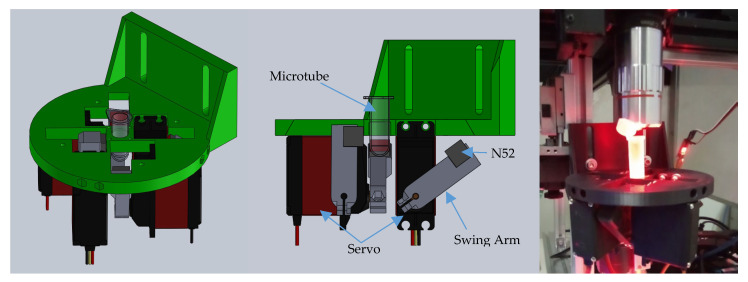
Design of Magnetic Actuator and Prototype.

**Figure 4 micromachines-15-00763-f004:**
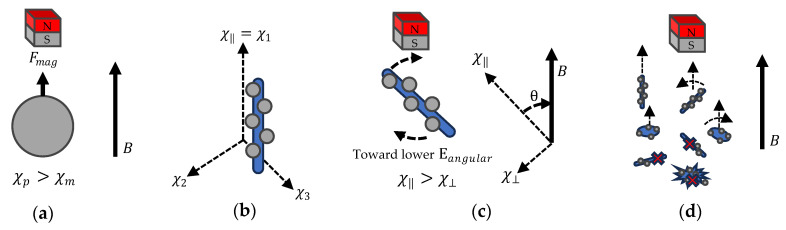
(**a**) Linear Magnetophoresis Force. (**b**) Anisotropic Magnetic Susceptibility. (**c**) Angular Magnetic Energy. (**d**) Decorated Particle Behavior in Magnetic Field.

**Figure 5 micromachines-15-00763-f005:**
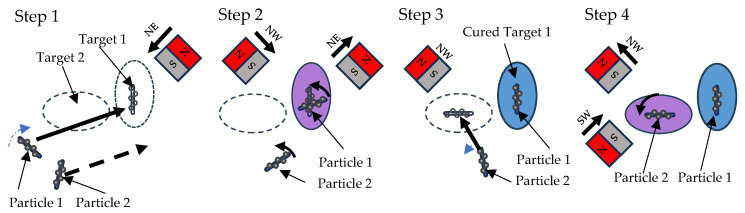
Particle Manipulation Scheme.

**Figure 6 micromachines-15-00763-f006:**
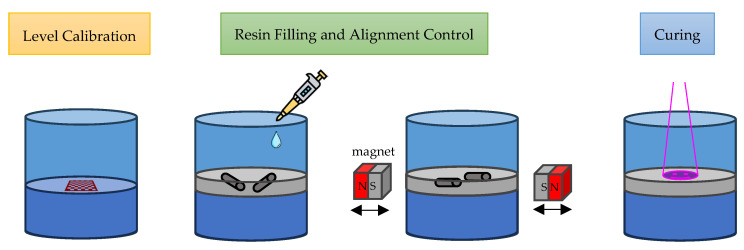
Anisotropic 3D Printing Procedures.

**Figure 7 micromachines-15-00763-f007:**
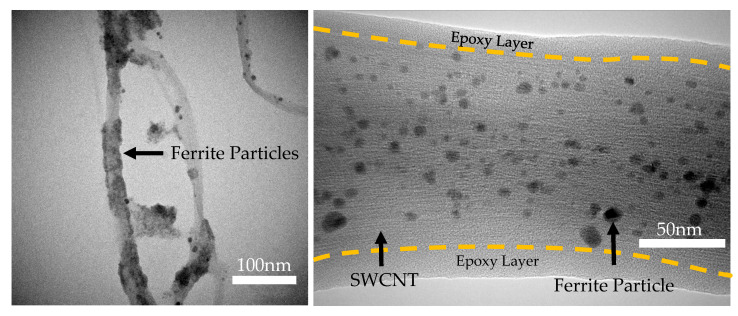
(**Left**) Ferrite@MWCNTs. (**Right**) Ferrite@SWCNT.

**Figure 8 micromachines-15-00763-f008:**
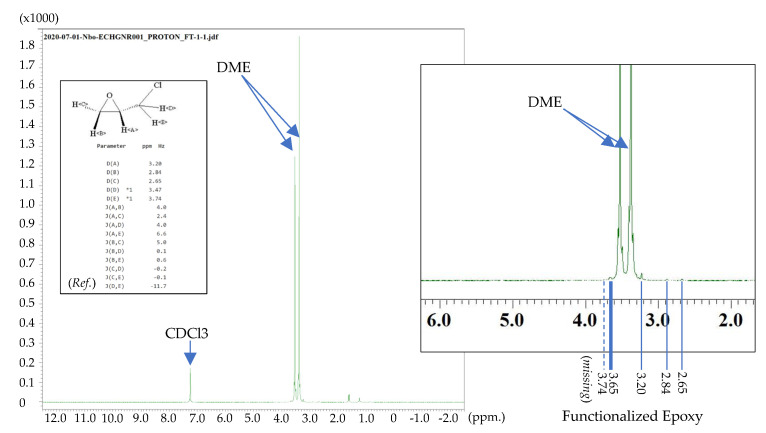
^1^H NMR Analysis of Epoxidized Carbon Nanomaterial (Ref.) [56].

**Figure 9 micromachines-15-00763-f009:**
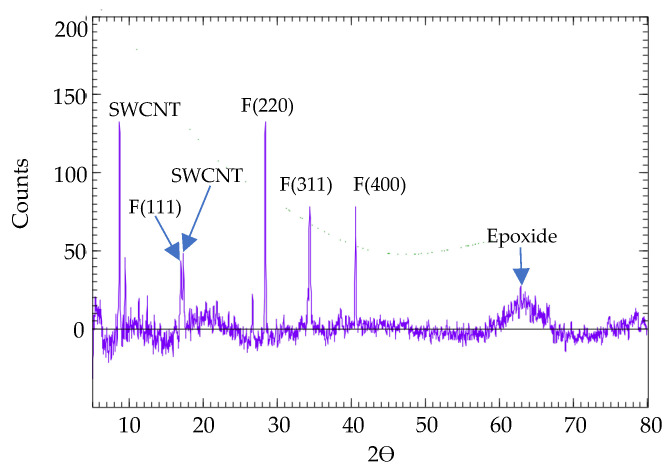
XRD Pattern of Ferrite-Decorated SWCNT.

**Figure 10 micromachines-15-00763-f010:**
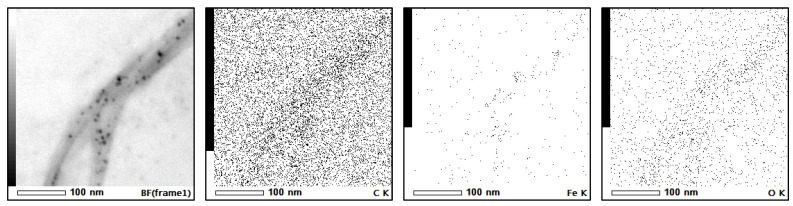
EDXS of Decorated SWCNT.

**Figure 11 micromachines-15-00763-f011:**
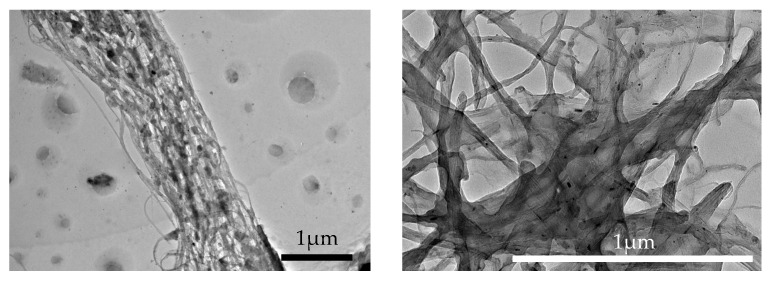
(**Left**) A Bundle of Decorated SWCNT. (**Right**) A Net of Multi-Layer GNRs.

**Figure 12 micromachines-15-00763-f012:**
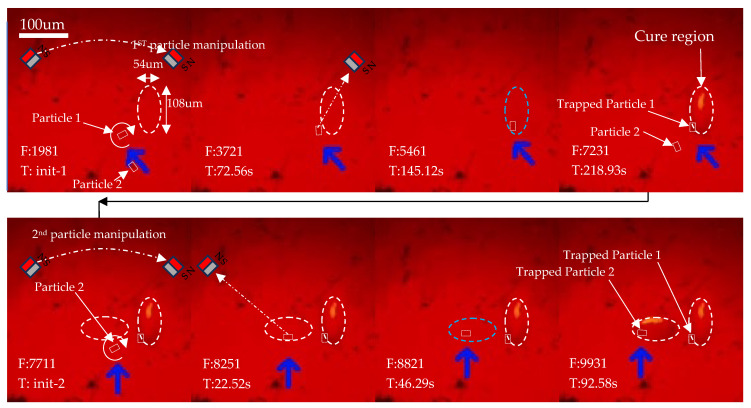
Alignment and Movement of Filament Bundle.

## Data Availability

The data presented in this study are available on request from the corresponding author.

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
