# Peer review of "Alignment Control of Ferrite-Decorated Nanocarbon Material for 3D Printing"

_micromachines, 2024, doi:10.3390/mi15060763_

Round 1

Reviewer 1 Report

Comments and Suggestions for Authors

The manuscript describes the procedure of preparing resigns, in which carbon materials
decorated with magnetc nanopartcles are incorporated. The applicaton of an external
magnetc feld aligns the magnetc nanopartcles, in the feld directon, which enables also the
alignment of the carbon material. Therefore, these resigns are considered as appropriate for
3D printng of specimens, with anisotropic propertes, due to the aligned carbon incorporatng
material. In general the idea is interestng. Nevertheless, there are several weak points, that
must be addressed: In partcular:
1. The introducton of the paper need to be more extensive, including further
informaton regarding the state-of-the art. In general, the anisotropic 3D printng
technology has already gained a lot of interest, thus there is huge amount of
corresponding literature. Furthermore it has to be furtherly (and more precisely)
explained why and how the anisotropic 3D printng technology is correlated to the
hereby study as well as why and how the current investgaton is innovatve in
comparison to the state-of-the-art.
2. The main aim of the paper should be clearly stated in both the abstract and the
introducton of the paper. As far as I understood, authors decorate carbon
nanoribbons with magnetc nanopartcles and then they use external magnets, in
order to align the nanoribbons. It is an interestng approach that it is not
highlighted neither in the abstract not in the introducton of the paper.
3. Synthesis sectons 2.1 and 2.2 must be extended, including more informaton, not
just refer to previous studies.
4. Line 84: “… using FeCl2 and FeCl3 as precursors.” Do they both used simultaneously
or separately? Authors must clarify.
5. A secton describing both the microscopic as well as the XRD characterizaton of
the studied samples should be incorporated, in the Materials and Methods part of
the paper.
6. Figure 1: Several steps of carbon materials decoratons is shown. It will be effectve
to add a short descripton to each step, either into the Figure or in the
corresponding capture. Furthermore in the fnal step Fe3O4 appears, however
there is not any descripton how it is made, in the corresponding secton. In
additon it becomes unclear how the Fe3O4 molecules are incorporated on the
carbon nanoribbon. Also it is unclear whether each ribbon has one Fe3O4
molecule or more, or it is cannot be counted.
7. Figure 3: It should be improved by labelling the corresponding parts, in order to
understand the basic components of the structure.
8. Figure 5: I did not fully understand the descripton of this fgure (lines 185-195). It
is a bit confusing. Authors may give a simpler descripton.
9. Figure 8: XRD peaks should be labelled according to their origin, either coming
from the Fe3O4, or from the carbon material.
10. Figure 11: I see nothing into the circles and the rectangles. What exactly should it
be shown;

11. Lines 153-160: The explanaton, regarding how the uniaxial magnetc propertes
arise is a bit confusing. Authors should give a simpler explanaton, which would be
understandable from people that are not familiar with magnetsm.
12. Line 82: “… as shown…” instead of “...as show…”
13. Line 91: “…GMA…” acronym should be explained when appears for frst tme, into
the manuscript. It is the same case with acronyms in lines 93-95, as well.
In general, as I said in the beginning, the current study is interestng, Nonetheless, I would
also expect to see a macroscopic inspecton of a property (i.e. the magnetzaton, or the tensile
strength) and how it is affected by the anisotropic 3D printng process. However this is just my
expectaton, which might be away from what authors would like to show in this study.

Author Response

Thank you for the comments and suggestions. I would like to apologize for the incorrectness and unclarity in this literature. 

  1. In this work, we try to integrate starting from the chemistry part through mechanical and electromagnetics problems. By doing so, previous work they did aim to manipulate a graphene/carbon nanotube and focus on building the machine superlatively. In contrast the integration in this work shows many other problems and more potential in this technology, such as the use of epichlorohydrin as the coupling agent and integrating the low-cost microscale as well as the magnetics system.
  2. We have included more elaborations on the abstract as well as in the introduction part as your suggesting and including a graphical abstract.
  3. We have extended the synthesis part as you recommended. I apologize for a short description and we have move some information in the appendix in to the section 2.1 and 2.2 for more completeness.
  4. Yes, and we have rewritten for more clarification.
  5. we have added more details of the instrument on the experiment in the appendix, since we want to balance between the information of the chemistry , optics and magnetics manipulation, therefore we try to avoid to including the introduction.
  6. We have extended this section as you suggested to explain the crystallization mechanism of the Fe3O4
  7. Figure 3 has been improved according to you suggestion.
  8. We have given a step number in the paragraph, which according to the Figure 5, for more clarification.

  9. Figure 8 has been labeled as your suggestion.
  10. The image sequence might not be well depicted. However, we have added more image figure 12 which have more magnification to show the result. For figure 11, we remain the same for illustrated the moving of the object that can be easier to observed on the VDO clips.

    And or the magnification of the microscope setup, we cannot change the objective lens to increase the zooming due to the lens also have a role in projecting a UV light, which changing lens will change a depth of field and the more magnification lens (10X), the depth of field would not be sufficient to cure a high depth of resin in the vat. Therefore, the current magnification is the optimum operation setup.

    However, I wish to have more improvement on this, but my Ph.D. program would not give me more time to investigate.
  11. The explanation of magnetic anisotropy is challenging. Most of the papers are best describing this phenomenon using the Energy perspective, which is the same approach as this paper. There are several phenomena interplay between the magnetic field and the material, which result in the total potential energy difference, that yields the rotational movement. Specifically explaining each phenomenon would be outside the scope of this work and make the reader more confusion. Therefore, we believe this is the best way to explain this phenomenon without going deep into those details.

  12. Thank you, they were corrected.
  13.  I apologize for this mistake. They have been corrected.

    Overall, I am grateful for the reviews and comments. Thank you.

Reviewer 2 Report

Comments and Suggestions for Authors

In this manuscript titled “Alignment Controlled of Ferrite Decorated Nanocarbon Material for 3D Printing”, ferrite-decorated graphene and carbon nanotubes is synthesized and applied to 3D printing. However, the materials ferrite and carbon Ferrite were no innovation. And the methods of magnetic field control and 3D printing, showed no significant advantage and advancement. In a whole, the paper cannot be accepted due to lack of novelty.

Other comments:

(1)     The structure and morphology of ferrite and carbon nanotubes are lacking, and only XRD and TEM data are available in this paper, which need to be further supplemented.

(2)     The EDS scanning image of O element in Figure 9 has low contrast and is not clear.

(3)     In the first three photos in Figure 11, the filaments mentioned in the paper cannot be observed. As important evidence in the paper, this figure should be re-collected.

(4)     The full name of TPO, PI-784, DLP should be given in the paper when mentioned for the first time.

Comments on the Quality of English Language

Moderate editing of English language required.

Author Response

Thank you for your comments. I apologize for the mistake I have sent in the previous version. For the novelty of this work, I do understand that in each field of research, such as in chemistry and micromachines. In chemistry, the decorated carbon nanotubes and unzipping of the graphene might exist for sometimes. Therefore, we have introduced Epichlorohydrin as a new technique to coupling and starting the crystallization of Fe4O4, this is one of our small focuses. The second work is to apply this material into microscale 3D printing. This microscale 3D printing might not be the latest innovation as well, they existed for quite sometimes.

However, integrate between these technologies pose and show more challenges which we could like to account in this work.

So, I would like to ask for the chance to publish my Ph.D work on this topic.  For the comments, we have corrected and elaborated according to you suggest and comment as follow,

  1. We have tried several more analyses, such as FTIR and NMR. However, the result could not show any evidence.
    For FTIR, since the CNT has a black color and the concentration of the ferrite nanoparticles is very low compared with CNT, therefore we cannot expect any readout from both ATR and transmissive. However, the TEM/EDSX will capture the existence of the decoration and give better evidence than what can we expect from FTIR.
    For NMR, since it is a magnetic material, it is prohibited to put into the machine. It could damage the NMR magnetic field and cause a problem to the facility. Anyway, we also have tried to put the sample in NMR in the process of coupling the epichlorohydrin to the nanotube. But we cannot expect any result from H-1 NMR since the reaction has a little involved with the hydrogen atom.

    Therefore, in this work we try to balance the information on chemistry synthesis as well as the other parts and we want to demonstrate the combination of this technologies that lead to the new advancement for many applications. We hope that this report can be picked and extend or gain deeper research that improves this method.

  2. Thank you, and we have changed the picture to yellow. We also point out some clarification of this image as well. The O image shows that if there is oxygen in the epoxide molecules covers all over the surface of the nanotubes, which does not exist in the raw carbon nanotube material. Therefore, O image confirms the grafting of epichlorohydrin. In the high vacuum stage of TEM, epichlorohydrin is easy to vaporize as well as the residue oxygen, so only the oxygen that is left on the sample should be in some kind of bond with the CNT. 

  3. I do agree that the filament is not easy to observe. However, Figure 11 is intended to depict the overall moving scheme. I would be clearer in the VDO. We have added Figure 12 to provide more Hi-res evidence on this work.

    We have tried to change the lens to 10X and 20X. But since higher magnification lens will have a shorter depth of field and the objective lens are also used for projection. Hence, the depth of cure of the resin is affected and not working well. For the CMOS sensor in this work, we have developed a polarization camera for observed some other polymer polymerization reaction such as polystyrene which have some polarization changed while it reacted. This phenomenon wasn’t well observed in our report recipe. So, we didn’t include, and we have no other suitable camera to replace. We also ran out of time and budget, so I would like to apology to this data collection.
  4. They are corrected.

    Overall, I am grateful for the reviews and comments. Thank you.

Reviewer 3 Report

Comments and Suggestions for Authors

The paper seems like an interesting attempt to align CNTs in resin with magnetic fields and may be of interest to people developing 3D printing methods. The authors may like to consider some of the minor points below:

Explain what DLP is in the introduction.

In Fig. 3 it would be helpful to provide an explanation of how the magnet is moved into position and a diagram or inset showing the magnet arm away and in position. A diagram of magnetic field lines relative to the printing area/volume may also be helpful.

Did any of the CNT materials have residual internal catalyst that was ferromagnetic?

Figure 7 RHS it is very difficult to see the SWCNT, not sure if more arrows or contrast can be altered?

Figure 11 It would be nice to have some optical images of the regions to show any orientation.

Author Response

Thank you for your comments, It give us a positive support.

We have add more explanation the DLP(tradename) into DMD(Digital micro device) in the introduction.

  • We do have a simulation from FEMM (a magnetics simulation). However, we didn’t include it in this work since the magnet line didn’t work only in a static field. The moving of the particle, especially for linear translation involves the oscillation or rotation of the magnetics field, which we are not well understood and there could be a lot more simulation and details regarding this phenomenon.

  • This is a very good observation. Anyway, in the experimental part, after the sodium solvation process, the process destroys any magnetic property. We have tested by putting the high strength magnet as well as it resided magnetic bar, and the solution didn’t show any more reaction toward the magnetics field. Therefore, we can conclude that the remaining catalyst has been neutralized.

  • We have altered the text as you suggestion.

  • We have added Figure 12 to show the high-resolution detail of the specimen.
  • Overall, I am grateful for the reviews and comments. Thank you.

Round 2

Reviewer 1 Report

Comments and Suggestions for Authors

Authors partially improved the manuscript according to the suggestions. However they do not gave sensible explanations to some of the comments. In particular: 

Comment 1: They do not include any information regarding the state-of-the -art, as suggested in my first comment. I still suggest the the introduction has to be extended in that direction. In addition, They just included 5 more lines (72-76) trying to explain the relation between the anisotropic 3D printing technology with the hereby investigation.

Comment 3: Section 2.2 was not extended as suggested. More information has to be incorporated in that section, not just in the appendix.

Comment 6: Regarding the comment, concerning Figure 1 description , the corresponding information, which is placed in the appendix, may be should be moved to the main text.

Comment 10: Regarding the comment, concerning the Figure 11, since “the image sequence is not well depicted” as authors state, then they could just properly rearrange the images to achieve better visualization. Figure 11 is a key figure, which demonstrates “the story”, which authors show in figures 4 and 5. In addition, Figure 12 is just a repetion of the down-right part of Figure 11. It does not add any further information to the text.

Comment 11: It is understandable the challenging nature of the magnetic anisotropy. However, hereby the anisotropy does not arise as an intrinsic property of the material. It is a consequence of the presence of magnetic particles around carbon ribbons. Therefore, no exceptional magnetic theory is needed to describe such a phenomenon. Thus I do not agree that “…this is the best way to explain this phenomenon without going deep into those details.”  as  authors state.

Considering the above I still believe that the manuscript must be furtherly improved, in order to fulfil the standards for publication.

Author Response

Thank you for your valuable comments on my manuscript. I apologize for any misunderstandings that may have arisen from my arguments. I appreciate the opportunity to clarify and improve my work based on your insightful feedback.

Regarding Comment 1,
As per your suggestion, we have rewritten the introduction to be more comprehensive and smooth, elaborating on the causality of the work.

Regarding Comment 3,
I apologize for misunderstanding the section. We have now extended Section 2.2 to include the reasons behind the selection of GMA and IBOA for the formulation. We hope this information enhances the completeness of the paper.

Regarding Comment 6,
The section has been moved as per your recommendation.

Regarding Comment 10,
We understand the complexity and have split the image into two steps for each manipulation of a filament. To clarify the manipulation of the magnetic field, we have added magnetic icons to represent the engagement of the magnets in manipulating the objects.

Regarding Comment 11,
I apologize for my previous argument. We have added a simple explanation of magnetophoresis. The intrinsic property I referred to is superparamagnetism, which occurs when a particle is approximately ten nanometers in size, which should be minimal in this work.

Thank you for your suggestions that have guided us in improving this document. We hope we have now achieved the standard required for publication.

Best regards,

Reviewer 2 Report

Comments and Suggestions for Authors

In this manuscript titled “Alignment Controlled of Ferrite Decorated Nanocarbon Material for 3D Printing”, ferrite-decorated graphene and carbon nanotubes is synthesized and applied to 3D printing. However, some revisions are required before publication and the following questions are suggested to be considered. 

(1)     The structure and morphology of ferrite and carbon nanotubes are lacking, and only XRD and TEM data are available in this paper, which need to be further supplemented.

(2)     The EDS scanning image of O element in Figure 9 is not clear and it is difficult to distinguish the shape of SWCNT.

Comments on the Quality of English Language

Minor editing of English language required

Author Response

Thank you for your comments on my manuscript.

  1. We have found the ^1H NMR analysis for this sample. As mentioned, we cannot put the ferrite-decorated sample in the NMR machine. However, we have located the NMR of the intermediate reaction, post-epichlorohydrin introduction. At the time of our sampling, since it had not reached the final encapsulation, the presence of DME as a solvent was still high to prevent agglomeration. Nevertheless, there are some peaks that confirm the grafting of epichlorohydrin. We also attempted ^13C NMR, but the noise was too high to provide conclusive evidence.

  2. We have replaced the TEM/EDXS image set. This new set shows more contrast in the oxygen image. However, since the SWCNTs were split, they became thin and only a few remained. Therefore, the contrast in the carbon image is reduced, and the background carbon atoms from the polymeric TEM grid are present.

I hope these corrections satisfy your comments. Additionally, following feedback from other reviewers, we have rewritten the introduction to be smoother and more detailed. We have also reorganized the experimental section and provided more explanations in each crucial part.

I hope this revision meets the requirements for publication.

Best regards,

Round 3

Reviewer 1 Report

Comments and Suggestions for Authors

Authors did revise the manuscript this time, and improved its overall presence.